

3 **An evaluation of short Earth Science CPD for trainee primary school teachers.**

5                                     Denise Balmer

7                    Department of Assessment, Curriculum and Pedagogy

8                    Institute of Education, University College of London

9                        20 Bedford Way, London WC1H 0 AL

10                 Correspondence to denise.balmer1@ntlworld.com

**Abstract**
The article is derived from a PhD thesis investigating the potential of earth science for the development
of primary school science. The evaluation from workshops run by the Earth Science Education Unit for
trainee primary teachers was appraised to assess the effectiveness of the short CPD programmes over
the period 2009-2015. Trainee teacher comments are analysed using thematic analysis which identified
points identified by Guskey (2000) as being the most important ideas for effective CPD programmes.
Despite these workshops being short, lasting generally less than two hours each, the conclusion
reached was that they offered useful teaching ideas, resources and background information which the
trainees could and would apply in the classroom.





**An evaluation of short Earth Science CPD for trainee primary school teachers.**
## Introduction
The Earth Science Education Unit was founded as a pilot scheme in 1999, and rolled out across the
United Kingdom in 2002, to encourage and enhance earth science teaching by both primary and
secondary teachers. The Unit was based at Keele University under the auspices of Professor Chris
King and initially sponsored for some 15 years by UK Oil and Gas. Earth Science CPD sessions which
delivered the requirements of the National Curriculum and beyond, were presented by a group of
trained volunteers, themselves earth scientists, who offered enthusiastic and accurate information and
methodology using low cost resources. Evaluation of the secondary programme was carried out in 2009
(Lydon & King, 2009) but the primary teachers' programme has only recently been examined. The
programmes given to trainee primary teachers over the period 2009-2015 were thoroughly assessed in
2018. The workshops had been revised in 2014 to comply with updates in the primary science
curriculum. The following article is derived from my PhD thesis which examined the potential of earth
science for the development of primary school science.

Ofsted (2013) stated that where primary science teachers and science leaders had received subject-
specific science CPD sessions, primary science teaching was more effective; in Ofsted's words "more
likely to be outstanding". Australian primary science teachers affirmed that short (up to four-hour long)
CPD workshops increased their self-efficacy and had a positive influence on their science teaching
(McKinnon & Lamberts, 2014). However, previously Adey et al., (2004) had suggested that the only
short CPD courses that would have any real impact on teaching would need to be very specific,
perhaps on software applications or assessment methods. The Wellcome Trust report (2013) found that
where science subject leaders had received science CPD they could better help any primary teacher in
their school who was struggling with science. Shallcross et al., (2010) suggested there was a need for
good integrated science CPD which included background information as well as specific-subject
knowledge and pedagogy. Abrahams et al., (2012) also felt that there was a need for CPD, especially
for practical work which they thought did not always have clear objectives but was often used to provide
a 'fun' lesson. They felt there was a need to make practical work more effective, and their Getting
Practical CPD programme was designed to support practical work in science. There has been little
published research on the effectiveness of primary science CPD programmes to date. Many local
teachers in my county have been disappointed at the lack of actual science knowledge and application
available at so-called primary science CPD which has seemingly concentrated mainly on pedagogy.
Primary teacher training establishments too, concentrate more on the pedagogy of teaching science
rather than actual information, which given that most primary trainees (and teachers) are non-scientists
is disappointing, (Wellcome Trust, 2013).
The primary earth science workshops I taught were specifically designed to meet the needs of non-
science primary teachers. Evaluation of the secondary ESEU workshop data by Lydon and King (2009)
showed that this CPD gave teachers both subject content knowledge and pedagogical knowledge,
increasing their confidence and effectiveness. Changes to most of these secondary teachers' teaching
methods were long term, as shown by a follow up survey carried out a year after the workshop (Lydon
& King, 2009). I analysed the ESEU data collected from the primary trainee teachers' evaluation forms
using thematic coding after the idea proposed by Braun and Clarke (2006). Some of Guskey's thoughts
of the range of experiences that teachers could be expected to benefit from a CPD were identified as



the themes from the collected data. These themes were the participants' reactions, their learning and
the proposed use of the new skills and knowledge gained from the CPD activities (Guskey, 2000).

## 1. Method of ESEU data collection from CPD primary workshops 2009-2015


The ESEU data were collected during trainee teacher workshops over the period 2009-2015. The
workshops were run in a wide range of primary teacher training institutions by their local ESEU-trained
facilitator. These various training institutions throughout England had requested a free primary earth
science workshop through Keele University. All workshop facilitators had been trained by the ESEU and
completed annual updating training, to keep them in touch with new concepts in earth science and
curriculum changes, particularly with the introduction of the new primary science curriculum in 2013.
The primary trainee teachers participating in the ESEU workshops were from a range of training
institutions across England and were on Teach First, PGCE or BAEd. courses or were on school
centred initial teacher training programmes (SCITT). The trainees' backgrounds and ages varied
greatly, some were British nationals, others were from overseas, these data do not show the
differences. The workshops comprise a series of low-cost, practical investigations and simulations
which can take place in any classroom and are each about 90 minutes long. In the workshops, the
participants were encouraged to work on as many of the investigations or simulations as they could, in
order to gain as much experience as possible during the time available. The facilitator worked with the
trainees, responding to theoretical and practical questions as they arose. The participants were asked
to evaluate the workshop sessions after they had taken part in them and the data and comments from
these evaluations, collected by the ESEU were made available for analysis. The evaluation form
requested background information about the trainee teacher's science and earth science training since
taking GCSE and whether the trainee teacher felt confident teaching earth science before the workshop
input. Given the large sample size, the evaluation forms used were the first 25% of forms completed for
each year, taken from the archive in the order they had been collected at Keele. This is not necessarily
the order in which the workshops were taught.
After completing the workshop, each participant was given the resource lists, risk assessments and
workshop instructions for the three primary workshops taught, so they could use the materials in their
schools immediately and pass the workshop information to their peers. The photograph shown in Figure
1 shows trainee teachers investigating soil.









**Figure 1 Trainee teachers investigating soil**

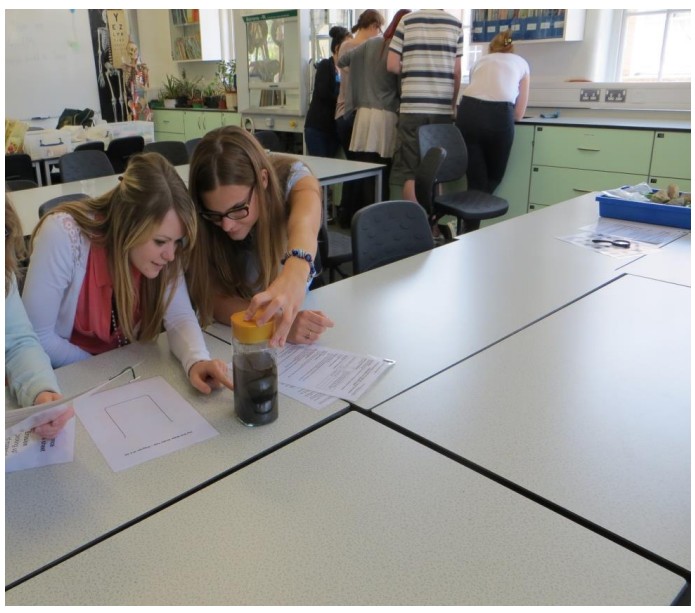


For the pilot study, I gathered data from 125 ESEU evaluation forms for each of the years 2009-2014,
this figure was based on 25% of the completed forms for 2009. This provided enough data for the pilot
(750 forms) but was not a true sample as the number of forms completed in each year was not the
same. I therefore increased the collected data to 25% of the evaluation forms for each year the
programme was taught, 2009 to 2015 (1395 forms). The ESEU data are partly in Likert scale form, but
the part of the evaluation of most interest to me was the 'comments section' written immediately after
the workshop. The ESEU evaluation form requested data in several formats:
• Background information on trainee teachers (these data have been used for the purposes of this
thesis)
• Eleven questions to be answered on a Likert scale referring to amount of earth science that
trainees may be teaching (most of these data were not used for the purpose of this thesis)
• Participants' comments about their workshop experience (these data have been used for the
purpose of this thesis).

When analysing these data, I transcribed all the comments on the sampled evaluation sheets for
determining themes in order to be able to analyse them using thematic analysis (Braun & Clarke, 2006).

## 2 Results of the ESEU data collection: The data

The background information data were extracted from the evaluation forms and tabulated so that
different years could be compared as seen in Table 1.
From Table1 it can be seen that the number of female trainees participating in the workshops is much
greater than the number of male participants, who are barely one-fifth of the overall total, in line with
Government statistics for 2015 which show that 85% of primary teachers are female (DfE, 2015 p7).



**Table 1 Compilation background data of primary trainee teachers taken from the data on the ESEU evaluation forms 2009-2015.**

|  | 2009 | 2010 | 2011 | 2012 | 2013 | 2014 | 2015 | totals | % of total |
|---|---|---|---|---|---|---|---|---|---|
| Total number of trainees in workshops in year | 424 | 452 | 688 | 1252 | 1196 | 1144 | 424 | 5580 |  |
| No. of evaluation forms used in study | 106 | 113 | 172 | 313 | 299 | 286 | 106 | 1395 | 25% |
| No. of females in study | 84 | 101 | 129 | 253 | 217 | 233 | 78 | 1095 | 78% |
| no. of males in study | 22 | 12 | 43 | 60 | 82 | 53 | 28 | 300 | 22% |
| Earth Science studied to 16 | 62 | 73 | 108 | 163 | 149 | 207 | 61 | 823 | 59% |
| Earth Science studied to 16+ | 13 | 9 | 15 | 29 | 21 | 26 | 8 | 121 | 8.70% |
| Earth science as minor part of degree | 17 | 8 | 15 | 39 | 26 | 26 | 3 | 134 | 9.70% |
| Earth Science as major part of degree | 9 | 5 | 5 | 4 | 13 | 2 | 1 | 39 | 2.80% |

The number of trainees who stated they had learnt any earth science or geology during GCSE was 59%. A small amount of earth science was included in GCSE physics/chemistry up to 2014, but the respondents may not have appreciated earth science as a specific topic within the curriculum. These workshops mostly took place before the 2014 changes in the National Curriculum which have now virtually removed earth science from the secondary science curriculum, placing it in geography with a more social emphasis, which means that the next generation of teacher trainee recruits will probably have studied even less earth science, from a science perspective, up to the age of 16. There is, however, more earth science in the primary curriculum from 2014. About 10% of trainees said they had studied earth science / geology after GCSE with some stating it was a minor part of a degree course (approximately 10%) whilst others had studied earth science as a larger part of their degree (2.8%). But overall, few primary trainee teachers in my sample have science degrees (Table 2), although it is not necessarily the case that those who do are able to teach science better than their colleagues as they sometimes cannot relate their science studies to the level required in primary school (PSST, 2016).

**Table 2 Number of trainee teachers with science degrees attending workshops**

|  | 2009 | 2010 | 2011 | 2012 | 2013 | 2014 | 2015 | Totals | % of total |
|---|---|---|---|---|---|---|---|---|---|
| Number of trainees participating: | 106 | 113 | 172 | 313 | 299 | 286 | 106 | 1395 |  |
| Degree in biology | 7 | 3 | 1 | 2 | 10 | 2 | 0 | 25 | 1.8% |
| Degree in chemistry | 0 | 1 | 0 | 1 | 1 | 2 | 0 | 5 | 0.40% |
| Degree in physics | 1 | 1 | 1 | 1 | 2 | 0 | 0 | 6 | 0.43% |
| Degree in earth science | 1 | 1 | 3 | 4 | 1 | 0 | 0 | 10 | 0.72% |
| Degree in geology | 0 | 0 | 0 | 0 | 0 | 0 | 0 | 0 | 0% |





Further data from the evaluation form is shown in Table 3 which shows trainees' confidence in teaching
primary science. (Note: some teachers were confident in more than one subject.

**Table 3 Actual numbers of primary trainee teachers who felt confident in teaching primary science**

| | 2009 | 2010 | 2011 | 2012 | 2013 | 2014 | 2015 | Totals | % of total |
|---|---|---|---|---|---|---|---|---|---|
| **Number of trainees participating:** | 106 | 113 | 172 | 313 | 299 | 286 | 106 | 1395 | |
| **Teaching confidence in biology** | 63 | 72 | 114 | 210 | 186 | 233 | 57 | 935 | 67% |
| **Teaching confidence in chemistry** | 16 | 16 | 20 | 25 | 32 | 36 | 30 | 175 | 13% |
| **Teaching confidence in physics** | 21 | 18 | 27 | 46 | 40 | 33 | 22 | 207 | 15% |
| **Teaching confidence in earth science** | 3 | 2 | 10 | 17 | 18 | 12 | 6 | 68 | 4.9% |
| **Teaching confidence in geology** | 2 | 0 | 0 | 0 | 0 | 3 | 0 | 5 | 0.40% |
| **Teaching confidencen all** | 1 | 0 | 3 | 3 | 3 | 1 | 2 | 13 | 0.93% |
| **No confidence** | 0 | 2 | 3 | 8 | 18 | 10 | 27 | 68 | 4.9% |


Since it is difficult to compare the raw data, Table 4 shows the same data transposed into percentages.
**Table 4 Percentage of trainee teachers who felt confident at teaching particular science subjects**

| | 2009 | 2010 | 2011 | 2012 | 2013 | 2014 | 2015 | Average % |
|---|---|---|---|---|---|---|---|---|
| **Number of trainees participating:** | 106 | 113 | 172 | 313 | 299 | 286 | 106 | |
| **Teaching confidence in biology** | 59 | 64 | 66 | 67 | 62 | 81 | 54 | 65% |
| **Teaching confidence in chemistry** | 15 | 14 | 12 | 8 | 11 | 13 | 28 | 14% |
| **Teaching confidence in physics** | 20 | 16 | 16 | 15 | 13 | 12 | 21 | 16% |
| **Teaching confidence in earth science** | 3 | 2 | 6 | 5 | 6 | 4 | 6 | 4.6% |
| **Teaching confidence in geology** | 2 | 0 | 0 | 0 | 0 | 3 | 1 | 0.85% |
| **Teaching confidence in all** | 1 | 0 | 2 | 1 | 1 | 1 | 2 | 1.1% |
| **No confidence** | 0 | 2 | 2 | 3 | 6 | 3 | 25 | 5.9% |


The data in Table 4 show that between 2009 and 2015, 65% of the participants stated they were
confident in teaching primary biology, but confidence in teaching chemistry, physics, earth science and
geology (the other sciences in the primary science curriculum) was much lower at 14%, 16%, 4.6% and
0.85% respectively. In 2015, however, confidence in teaching biology within the sample, had fallen from
a high the previous year, to its lowest level, whilst the same year, 2015, showed an increase in
confidence in teaching chemistry and physics. This difference between chemistry and physics, on the
one hand, and biology, on the other, may relate to the 2014 changes to the primary curriculum, which
reduced the amount of chemistry and physics in the curriculum. Overall, though, a much higher
percentage of teachers had no confidence in teaching primary science in 2015 (25%), a huge increase
on previous years, as seen in Figure 2.



**Figure 2 Bar graph showing overall trainee teacher confidence in teaching primary science from 2009-2015**

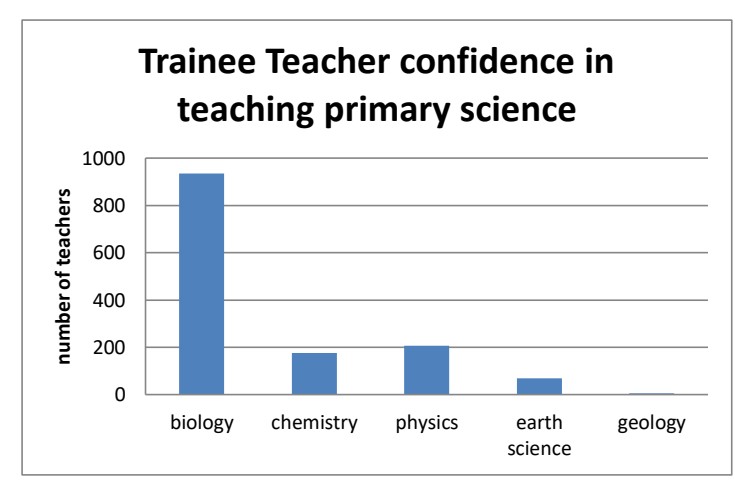

If teachers are not confident in their ability to teach a subject, this can often affect their enthusiasm and ability to enthuse their pupils (Aalderen-Smeets et al., 2013). Across the 2009-2015 period, only 1.1% of the trainees stated that they were confident at teaching all of primary science.

Confidence in teaching geology/earth science was low (averaging 5.7% across the 2009-2015 period) before the workshop, as stated by the trainees on the evaluation form (Figure 3).

**Figure 3 Percentage of teacher trainee participants at ESEU workshops stating they had no confidence in teaching primary science prior to participating in the workshop**.

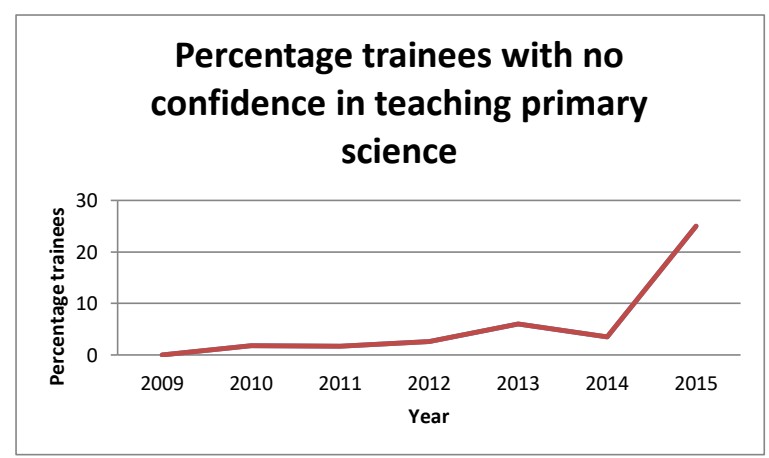

One worrying feature is that the graph suggests an increasing percentage of primary trainees who state they have no confidence in teaching primary science (Figure 3). Since the major increase occurs after





the implementation of the new National Curriculum it may be that trainees feel less confident with the
new programmes and their assessment procedures.
A Likert scale was used in the CPD evaluation form to ascertain whether the respondents felt the
workshop had increased their confidence. All participants indicated that their confidence had increased
and many of the comments used in the later analysis stated that their knowledge and understanding
had improved.

## 195 3.1 Trainee comments written on the ESEU evaluation forms

The trainees were asked to comment about their workshop experience on the evaluation form. There
were 2365 comments from the 1395 participants; these were transcribed and classified into six themes
in the following manner, as described by Braun and Clarke (2006). A list was made of all the comments
and these were initially grouped under headings (Table 5) which were then categorised to form themes.
These themes were identified as the main benefits the trainees had identified from the workshop: the
practical nature of the investigations and simulations; the engaging nature of the workshops; the
usefulness for their own future teaching; the simplicity and availability of the resources used in the
investigations and simulations; other positive points; and negative points. The numbers of comments
are listed by year and the themes to which they were allocated are shown in Table 5.
**Table 5 Composite table of comments and themes from participants about ESEU CPD workshops 2009-**
**2015**

| Comments from evaluation forms | Theme | 2009 | 2010 | 2011 | 2012 | 2013 | 2014 | 2015 | Total |
|---|---|---|---|---|---|---|---|---|---|
| Practical / Hands-on | 1 | 46 | 38 | 67 | 81 | 77 | 87 | 24 | 420 |
| Models | 1 | 0 | 4 | 0 | 0 | 1 | 1 | 0 | 6 |
| Good experiments | 1 | 2 | 4 | 4 | 20 | 19 | 14 | 20 | 83 |
| Interactive/investigative | 1 | 2 | 1 | 9 | 17 | 15 | 10 | 2 | 57 |
| Useful/valuable/effective | 1 | 10 | 0 | 18 | 40 | 20 | 50 | 1 | 139 |
| Interesting/good background | 2 | 15 | 4 | 12 | 40 | 16 | 0 | 18 | 105 |
| Engaging/enjoyable/fun | 2 | 23 | 12 | 36 | 39 | 42 | 27 | 9 | 188 |
| Fantastic/brilliant/excellent | 2 | 13 | 17 | 9 | 11 | 23 | 0 | 18 | 91 |
| Creative/inspiring/ | 2 | 0 | 6 | 0 | 2 | 0 | 5 | 0 | 13 |
| Presentation/ambience | 2 | 0 | 0 | 1 | 1 | 0 | 1 | 0 | 3 |
| Presenter's knowledge | 2 | 0 | 0 | 5 | 14 | 33 | 30 | 10 | 92 |
| Discussion /informal/experiences | 2 | 4 | 4 | 6 | 3 | 3 | 5 | 1 | 26 |
| Enthusiasm/passion for ES | 2 | 0 | 2 | 8 | 14 | 8 | 6 | 4 | 42 |
| Answered participants' questions | 2 | 0 | 1 | 2 | 5 | 2 | 6 | 2 | 18 |
| Great teaching ideas | 3 | 16 | 19 | 29 | 62 | 86 | 65 | 20 | 297 |
| Good information/concepts | 3 | 12 | 8 | 13 | 30 | 24 | 23 | 14 | 124 |
| Useful in class/lesson plans | 3 | 0 | 19 | 5 | 26 | 35 | 32 | 18 | 135 |
| Relevant to curriculum | 3 | 0 | 7 | 23 | 13 | 7 | 22 | 6 | 78 |
| Right level/easy instructions | 3 | 0 | 3 | 6 | 2 | 12 | 4 | 2 | 29 |
| Extensions | 3 | 0 | 0 | 1 | 1 | 0 | 0 | 0 | 2 |
| Adaptable | 3 | 0 | 1 | 1 | 4 | 2 | 0 | 0 | 8 |
| Differentiation | 3 | 0 | 0 | 0 | 2 | 4 | 0 | 0 | 6 |
| Good for SEN | 3 | 0 | 0 | 0 | 0 | 1 | 0 | 0 | 1 |
| Fits own teaching | 3 | 3 | 0 | 2 | 6 | 1 | 4 | 0 | 16 |
| Easy delivery | 3 | 8 | 0 | 1 | 2 | 0 | 0 | 0 | 11 |
| Good vocabulary | 3 | 2 | 1 | 1 | 0 | 0 | 2 | 0 | 6 |
| Gives confidence/deliverable | 3 | 2 | 9 | 3 | 18 | 11 | 8 | 5 | 56 |
| Cross curricula links | 3 | 3 | 0 | 1 | 0 | 2 | 3 | 0 | 9 |
| Misconceptions | 3 | 0 | 0 | 0 | 2 | 0 | 0 | 0 | 2 |
| Relates to real world | 3 | 0 | 0 | 0 | 0 | 4 | 3 | 3 | 10 |
| Correlates life skills | 3 | 2 | 1 | 0 | 0 | 0 | 0 | 1 | 4 |
| Improves thinking skills | 3 | 2 | 1 | 0 | 1 | 2 | 1 | 0 | 7 |
| Evokes curiosity/insightful | 3 | 0 | 0 | 2 | 1 | 0 | 4 | 0 | 7 |
| Improves understanding | 3 | 0 | 0 | 5 | 4 | 0 | 18 | 6 | 33 |
| Improves own knowledge | 3 | 10 | 0 | 0 | 0 | 0 | 4 | 1 | 15 |
| Useful resources | 4 | 18 | 15 | 9 | 14 | 27 | 26 | 11 | 120 |
| Good CD ROMs | 4 | 0 | 0 | 5 | 0 | 1 | 5 | 13 | 24 |
| Clear explanations | 4 | 6 | 0 | 0 | 0 | 0 | 3 | 0 | 9 |





| | | | | | | | | |
|---|---|---|---|---|---|---|---|---|
| **Knowledge giving/good info.** | 4 | 2 | 0 | 0 | 0 | 0 | 0 | 2 |
| **Not overloaded** | 5 | 3 | 0 | 1 | 4 | 0 | 2 | 0 | 10 |
| **Too short** | 5 | 1 | 0 | 3 | 16 | 1 | 7 | 2 | 30 |


The themes are identified below and shown as a graph in Figure 4.
• Theme 1 Practical: 705 comments relating to effectiveness of practical activities and investigations,
and the usefulness of the CPD in the classroom.
• Theme 2 Engaging: 578 participants' comments about how workshops would be received by
primary children and learning points which could be made.
• Theme 3 Teaching: 856 comments about the ease of delivery, use of good vocabulary,
differentiation uses, level of approach, clarity of explanations.
• Theme 4 Resources: 155 comments related to the simplicity, availability and inexpensive use of
everyday items for the investigations and simulations.
• Theme 5:30 positive comments including ones on length and timing of the CPD workshop, and how
the participants felt towards teaching earth science after the workshops.
• Theme 6: 41 negative comments including those from participants who did not intend to use the
exercises in their classes.

**Figure 4 Workshop theme analysis**

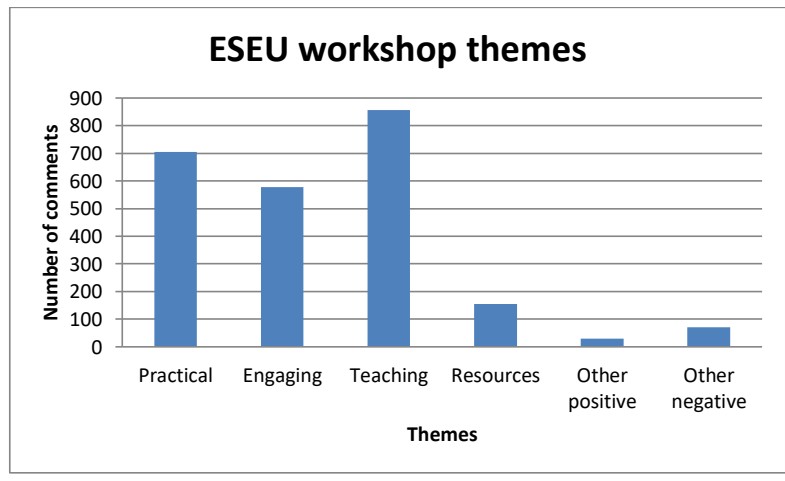


In the 'practical' theme, trainees' comments stated that the workshop sessions provided effective
simulations and hands-on practical investigations that were both interactive and investigative. Trainees
felt these investigations would appeal to the children's imagination and that pupils would identify with
the concepts from the investigations, thus dispelling alternative conceptions, evoking curiosity and
improving thinking skills and knowledge and understanding. This can be seen as effective pedagogy,
enabling learning. The workshops gave ideas for making a simple water-cycle model; practical activities
to show how soil erosion could be curtailed by vegetation; and using a piece of guttering to replicate a
river's flow, simulating relevant experiences that children may experience in their local area.



The 'engaging' theme brought together the trainees' comments about their feelings of working on the
earth science investigations and how they thought these investigations and simulations would run in
their primary classroom. They also commented that the investigations would provoke discussion and
the asking of many questions, again invoking effective learning pedagogy as children would recall the
practical side of the investigations and working together
The 'teaching' theme included points about the good vocabulary, the ease of delivery, and the fact that
the experiments could be differentiated for differing abilities. Using scientific language in an appropriate
setting was an important point made; children could visibly see evaporation and condensation in the
water cycle simulation, and permeability could be measured in the rock and soil investigations. Trainees
felt that they could use the workshop materials in their own teaching and use them for cross-curricular
purposes as well.
The 'resources' theme recognised that these investigations could be carried out using simple
equipment made from everyday items, for example, lemonade bottles and coffee filters. It also
acknowledged the usefulness of the CDROM which contained all the necessary investigative ideas and
risk assessments.
Some of the positive points raised were the clear explanations given by facilitators, and the fact that the
materials could easily be differentiated and also used for SEN work. The subject knowledge input was
appreciated as was the discussion which arose during the workshop, as all the facilitators would
endeavour to explain the scientific concepts behind some of the practical investigations and
simulations. Negative points that were made were on the length of the CPD (too short) and the need for
more KS1/EYFS resources, despite the CPD being advertised for KS2 trainees.
Overall, the feedback was positive with few negative comments. The comments received from the
trainees about the ESEU workshop were very encouraging and shows what a well-designed short CPD
session can achieve. Trainee Teacher comments on how they will use their newly gained knowledge
are shown in Figure 7.5.
**Figure 5 Comments on how the CPD will be used**

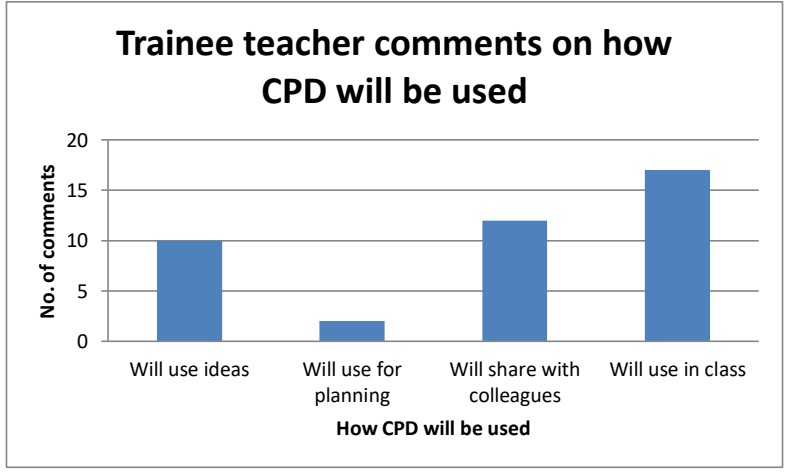




## 4. Identifiable pedagogy within the ESEU workshops


CPD of this nature can greatly enhance a trainee's pedagogical content knowledge by providing ideas
on how to teach concepts, increasing the trainees' self-efficacy and hence the likelihood that they would
use the material in their teaching. Various off-the-cuff comments from participants after a workshop
have been "Oh good, I have to teach soils/rocks in my next teaching practice, so now I know what to
do" and "I wish we had had this workshop before my last teaching practice as I had to teach about
rocks and soils and really did not understand it, but I do now".
The workshops offer opportunities for discussion and questioning, and for pupils to develop the
investigative ideas offered in different ways, to answer their own queries. For example, using the
investigation simulating coastal erosion, pupils can change the wave direction and strength, the size of
material being moved by the waves and the cliff material composition (more clayey, sandy, gravelly).
These different simulations can be linked to real life examples happening around the British coastline,
making them very relevant to where the children live or their holiday experiences. Learning becomes
more accessible and connected through noticing the changes in a practical manner, and children can
explain the erosion concepts from their observed understanding. Children give verbal feedback from
their visual experiences, and playing with sand and water has a 'wow' effect which may well be
remembered. All the investigations offered in the ESEU CPDs enable a range of concepts to be
examined and taught, which, when investigated at a simple level, applicable to the age of the
participants, provides a motivating and therefore hopefully lasting impression.
Trainees commented that providing concrete experiences using local resources would benefit their
teaching, as suggested by Fitzgerald (2012). The workshops continually promoted the use of local soils,
rocks and fossils and examples relating to the 'real world'. The simulations offered models to help
understand concepts such as the water cycle, a difficult idea for children to grasp. The CPD provides
effective teaching and learning as well as opportunities to assess children's progress through their oral
or written understanding.
The trainees identified ways that they would use their CPD session when in school. A number believed
they would be able to use the material directly, during teaching practice. Some also stated that they
would have liked to have had the resources and ideas earlier so they could have used them when on
teaching practice. Other trainees felt they could modify the ideas to fit their teaching programmes, whilst
others said they would share these ideas and use them for planning future work.
The themes categorised by the trainee teachers relate closely to those identified by Guskey (2000) as
being important outcomes for an effective CPD. Guskey suggested that CPD can be evaluated at five
levels of outcomes:
• level one: participant reactions
• level two: participant learning
• level three: organisational support and change
• level four: participants' use of new knowledge and skills
• level five: student learning outcomes.
Levels one, two and four are applicable here.





Level one, participant reactions, can be identified through all the positive and negative statements
made by the participants after the CPD (Table 7.5). Of the 49 different points identified, only four are
negative, showing that the statements made over the 2009-2015 period indicate positive reactions.
Level two, participant learning, is indicated within the themes in a number of places, not just under
'knowledge giving'. For example, comments such as 'good information given', 'answered participants'
questions', and 'discussion/informal experiences' all suggest learning.
Level four, participants' use of new knowledge and skills, has been graphed in Figure 7. 5 and identifies
how the participants say they will use the CPD information.
Since these were only trainee teachers participating in the CPD, they had no way of influencing their
organisations (level three) or of knowing student outcomes (level five) at the present time.
At the end of the workshop, each primary trainee was given a USB stick, which held a complete set of
the materials and instructions used in the workshop, linked to references in KS2 primary science
curriculum. This gave rise to the following comments: that the instructions had "clear explanations"; the
activities were "instantly available to use in the classroom because of the ease of obtaining resources";
and they gave "good knowledge in a format useful for children and trainees".

## 319  5. Discussion of the ESEU CPD results

The results from the analysis of the comments show that participants' feelings towards the workshops
were overwhelmingly positive with very few negative comments (1.7%). The CPD provided subject
content knowledge (SCK) and the pedagogical content knowledge (PCK) for teaching earth science for
trainees with little or no science background, enabling them to use scientific ideas confidently. Trainees
stated that the provision of resource materials such as the CDROM, which contained all the
investigations and risk assessments would be very useful when teaching this section of the primary
science curriculum. Informal discussion revealed that trainees were thinking further than the given
ideas, and in fact using the CPD as a starting point for other topics in the primary curriculum; for
example, the simulations of coastal erosion, river processes and water cycle can be linked to
geography, history, biology, design and technology. This makes the time spent on one CPD time well
used.
The main themes identified by the participants – practical, engaging, teaching and resources – all relate
to sound pedagogical practices as identified in the ten TLRP principles of effective pedagogy (James &
Pollard, 2011). The theme 'practical' embraces interactive, investigative practices, which are valuable
and effective. The trainee teachers were motivated and stated under the engaging theme that there
was scope for questioning and discussion leading to higher thinking and critical thinking. The' teaching'
theme entailed identifying misconception, use of appropriate vocabulary, adaptability and differentiation
activities, evoking curiosity and insightfulness, as well as being suitable for planning and later
assessment.
As already suggested the workshop identifies with those points identified by Guskey as being effective
CPD outcome levels. The CPD is therefore seen to be an effective teaching strategy in in its design and
delivery by its participants, providing an applicable short workshop when using Guskey's criteria.
A further piece of research which looked at the impact of focused CPD on teachers' subject and
pedagogical knowledge was undertaken by Scott et al (2010). These researchers stated that where
CPD was domain-specific and teachers were able to focus on learning, teachers found the CPD



effective and useful. Many respondents in this survey said that they would use the pedagogical ideas in
their teaching and that the CPD had provided additional subject content knowledge they could use.
Scott et al (2010) looked specifically at secondary physics and chemistry short CPD provision, because
of the shortage of secondary physical science teachers. King and Thomas (2012) evaluated short earth
science CPD intervention workshops for secondary teachers with similar conclusions. My research
suggests that these primary earth science CPD workshops were as effective as these secondary
workshops in providing both pedagogical and subject content knowledge.
The ESEU primary teacher trainee evaluation forms had not previously been investigated although
analysis of the CPD impact on secondary science teachers and science trainee teachers had been
undertaken (Lydon & King, 2009). That analysis of the secondary CPD showed that even though some
of the research literature concludes that short-term CPD is not effective, the ESEU CPD led to
increases in knowledge and understanding, at least as stated by the participants. Further, a follow-up
postal survey of participating secondary teachers carried out a year after the CPD indicated that
teacher practices had changed, indicating long-term benefits from these short CPD workshops (Lydon
& King, 2009).
The findings from the primary evaluation forms indicate that the workshops given to primary teacher
trainees were well received. Comments suggest that the trainee teachers intended to use earth science
in their primary science work because they saw it as being relevant to their pupils' everyday lives. King
and Thomas (2012) calculated the impact secondary ESEU short CPD workshops had on the number
of trainee teachers, teachers and, using a multiplier gauge, number of students. My research shows
how the primary education sector benefitted too, with some 700 primary teachers attending workshops
between 2008-2011 (ESEU data), who could influence some 18,000 primary pupils annually. The total
number of trainee teachers who had attended the workshops between 2009 and 2015 was 5580 (ESEU
data). The large majority of these trainees would be teaching pupils in the coming years, adding to the
number benefitting from the CPD.
The trainee primary teachers said that the materials fitted in well with their approach to teaching and
were relevant to the curriculum. Harlen and Elstgeest (1992) stated that it is important that teachers
have their own understanding of a subject before they teach it, or explain it to their colleagues. These
workshops provide that understanding at an appropriate level for primary science. Unfortunately, it was
not possible to follow up with a postal survey of the trainees' teaching practices, as was done for the
secondary workshops, since the trainees completed the activities whilst not in permanent employment
in schools, the time that has elapsed since the training took place is too great, and contact details are
not available.
Overall, the evaluation from these workshops suggests that the trainee teachers will use the materials
to the benefit of their primary pupils with confidence. This evaluation shows that the workshops are
fulfilling a need, by offering relevant subject and pedagogical knowledge and do increase confidence in
teaching primary science. The trainees were devising their own plans for implementing these
investigations, which will surely enrich their teaching, not just in earth science but by relating the
concepts they had learnt to the overall science curriculum.

## 6. Potential of earth science for the development of primary science
It is interesting that in the data the only science subject many of the primary teacher trainees felt
confident about teaching was biology, before participating in the CPD workshops. Perhaps biology is as
close as primary and secondary school science gets to looking at science which is relevant to young
people? Everyone has some understanding of their own biology, but we don't develop the science that





is around us all the time. The physics strand of the primary science curriculum is often seen as difficult
by trainee teachers, who feel less confident when having to teach it (McCrory & Worthington, 2018).
Earth science can be used to introduce physics concepts such as forces, using children's relevant
experiences of wind and its effects. In 2012 King suggested that Earth Science should not only form a
significant part of *primary* children's science curriculum but for *all* those children up to age sixteen.
Although the present primary science curriculum has included more earth science the linkages are
unclear and, as with the rest of this curriculum, topics are isolated where they could be so easily
integrated. Why are we not making greater use of earth science everyday materials and events in our
primary science teaching, as these are available resources of which we all have experience?
Every child needs to understand their own surroundings and how soils, rocks, weather plants and
habitats work together. Surely a better understanding of our own earth science would encourage
appreciation of the importance of local changes on a world scale. Now is the time to ensure the next
generation have this knowledge and understanding.

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

London.

Record 854373 An evaluation of short earth science CPD for trainee primary teachers: logged in
reshare@ukdataservices.ac.uk