# Peer review of "An evaluation of short Earth Science CPD for trainee primary school teachers."

_Geoscience Communication, 2020_

## Referee Comment (RC1) · Anonymous Referee #1 · 15 Jul 2020

This is potentially a useful article. Unfortunately, it looks as thou the author has rushed their submission. It needs a lot of straightforward tidying up by the author before a revised version is resubmitted.

Specific feedback:

1. Title is a bit unexciting - ? something like 'The value of short Earth Science CPD for trainee primary school teachers'.
2. Line 31. Not clear whether "but the primary teachers' programme has only recently been examined" refers to another study (in which case it should be referenced) or this study (in which case 'only recently is unnecessary and confusing).
3. Terms that may not be familiar to a non-English audience need explaining – e.g. 'Ofsted', 'Teach First', 'PGCE' and 'BAEd.'. Similarly, phrases like 'non-science primary teachers' (lines 57-58) may not be understood by someone outside England.
4. Don't use phrases like 'my county' – stipulate 'England' or 'the UK' as appropriate.
5. Guskey is introduced without any explanation.
6. There seems to be no mention of ethical issues.
7. Some passages look to have been inappropriately copied and pasted from a PhD thesis – e.g. 'these data have been used for the purposes of this thesis' (lines 112-113 – and similarly in the next two bullet points).
8. Some passages have not been well proof-read / laid out, e.g. in table 1 we have 'no. of males in study' instead of 'No. of males in study', 'totals' is in the wrong position and there is no need for the three occasions when a terminal zero is used in the percentages.
9. I don't think Table 3 is needed, given Table 4.
10. Figure 2 is clearly the wrong Figure and does not look to merit inclusion.
11. Figure 3 is what is referred to in the text as Figure 2.
12. The references need quite a bit of tidying up.

---

## Referee Comment (RC2) · Anonymous Referee #2 · 22 Jul 2020

General comments

An interesting informative paper. Or is it meant to be an article?

I'd recommend minor revisions. The content may well be of interest to the readership

Specific comments

The presentation of findings and analysis seem disjointed where reorganising and changing table 4 to incorporate the findings of the themes would make easier reading of these data

A summary of the earth science across the curriculum of the primary stage in England (KS1 and KS2) would be beneficial in a table so readers may understand the concepts

expected to be delivered.

Technical comments

Affiliation: It is the Department of Curriculum, Pedagogy and Assessment, CPA.

And it should say UCL IOE., not the other way around, that is the style meant to be used Abstract

Is it not a paper?

And surely not derived but could just say the paper investigates . . ..

Page 2 line 3 auspices. Does not sound appropriate. Surely leadership?

Line 27. Some 15 years is vague. Is there not a start and finish date?

Line 34 is it a paper or an article, I am unfamiliar with this publication. The It examines the potential. . .. Which was part of a larger study (Balmer, D. 2019) and put in refs and have this in the references?

Line 50. This part is very vague, inset one example is. . .. . .

Line 53

Is this true? Pedagogy is part of delivering information. It is useful to know the level of the understanding of the 4 sciences of these teachers before the 'treatment' was this done at the start as part of ESEU protocol? I would remove the sentence 'Many local. To pedagogy. It is too bland. The next sentence reported by Welcome trust stands. Perhaps a consideration of the disjointed way in which earth science is scattered amongst the three science traditionally taught in England. when the participants were probably at school Could be mentioned.

Line 58. Non-science background?

Line 62 The data analysis is in methodology, not here, thus delete sentence

Page 3. Line 71. In Southern England?

Page 4. Line 105 Pilot study? Surely this is irrelevant. It is the, line 303. Why?

Page 13 e data she collected, and which is reported.

Line 118 more information about the analysis here. How were the determining themes acquired, read, and a reducing of categories??

Line 124. this could put more authoritatively. The analysed data show (table 1) number of . . . and Line 125. Barley 1/5th. insert the % here

page 6 line 163. a significant increase? (by what?)

Page 8. Line 1i98. Again, and example of the categories I it seems a little disjointed t have the methodology mentioned here again but I am not familiar with the format of papers in this platform. AH they appear on page 9, 211-223 or have them in the tb ale with constituent members underneath the major themes heading, would be much easier to read. Now are the themes derived from Table 5. So, take 5 would be altered with these lines before the table.

Page 9. Line 242. Good as opposed to bad or evil vocabulary surely author means appropriate and relevant to years groups

P 10 line. 257. On first use of these initials they need writing out. non-English readers may not be familiar, like wise after saying what ESEU is the acronym should be KS 1. And other refs Page 13 line387

As. Former Ofsted inspector and biologist I dispute this. A few facts are delivered in bytes! I agree with the authors further comments!

---

## Author Comment (AC1) · 26 Sep 2020

I thank the two referees for the time and consideration they have put into reading my paper, and their constructive criticism. I believe I have taken on board their comments and made appropriate amendments throughout.

Response to Referee 2 My apologies for getting the affiliation incorrect, this has now been amended. Line 3 has been amended, and a start/finish date included at line 27. The section at line 34 has been rewritten as has the section around line 53 with the inclusion of where earth science topics are located in the curriculum at lines 34-40. Line 59 background has been replaced with term non-scientists and text rewritten. Comment re line 62 not understood as the phrase is within the introduction. Line 71

[Figure]

– in England stands. The workshops took place throughout England during 2009-2015. Line 105 Reference to the pilot study has been removed. Line 118. Extra detail has been inserted Line 198 Extra detail inserted Line 242 Good replaced with word 'useful' Line 257 The referee is entitled to their opinion. I can only say that my 40 years of experience working in schools with primary teachers has given me insight into how many of them feel about teaching science. I have worked with teachers and trainees in the north and south of England. I believe have made amendments and added information as suggested throughout, and hope the paper is now acceptable. (Dr) Denise Balmer

---

## Author Comment (AC2) · 8 Oct 2020

I thank the two referees for the time and consideration they have put into reading my paper, and their constructive criticism. I believe I have taken on board their comments and made appropriate amendments throughout.

Response to Referee 1 1 The title has been amended as suggested 2 The sentence has been reworded, 'The programmes given to trainee primary teachers over the period 2009-2015 were thus thoroughly assessed in 2018.' 3 All the terms used were explained and sentence rephrased: 'The primary earth science workshops I taught were specifically designed to meet the needs of primary teachers with non-science backgrounds'. The primary trainee teachers participating in the ESEU workshops were

from a range of training institutions and programmes across England. Four different teacher training programmes were available during this period: Teach First: a programme where participants work in schools and are fully paid whilst on a two-year training course. The trainees, who have a wide range of backgrounds and experience are supported by tutors and day release sessions Post Graduate Certificate of Education (PGCE) Bachelor of Education (BAEd) courses SCITT courses: school centred initial teacher training programmes 4. 'my county' replaced as suggested 5 Guskey's ideas explained 6 Ethical issues: permission was given by trainees in the photograph for use of photograph. Evaluation forms asked for permission to use comments and only those forms with permission were used. 7 Corrections made 8 Corrections made 9 Table 3 deleted and adjustments made in text 10 Figure 2 shows data from Table 4 (now Table 3). 11 the references have been tidied up ! I believe have made amendments and added information as suggested throughout, and hope the paper is now acceptable. (Dr) Denise Balmer

---

## Author Response (AR2)

Dear Editor

I have made the minor revisions you recommended and am sending both a word copy and pdf as my copy looks alright when I review it, but I know I had difficulty previously. I have also tried to ensure tables are situated on a single page, but have had difficulty with this too.

I only had access to the evaluation forms for three days, and therefore do not have full quotes from respondees, but short sentences or just words - I hope what I have included is suitable.

Thank you for your continued support

Denise Balmer

[revised manuscript text omitted]